# Unveiling Sustainable Potential: A Life Cycle Assessment of Plant–Fiber Composite Microcellular Foam Molded Automotive Components

**DOI:** 10.3390/ma16144952

**Published:** 2023-07-11

**Authors:** Tao Feng, Wei Guo, Wei Li, Zhenghua Meng, Yao Zhu, Feng Zhao, Weicheng Liang

**Affiliations:** 1Hubei Key Laboratory of Advanced Technology for Automotive Components, Wuhan University of Technology, Wuhan 430070, Chinazhmwhut@126.com (Z.M.);; 2Hubei Collaborative Innovation Center for Automotive Components Technology, Wuhan University of Technology, Wuhan 430070, China; 3Hubei Research Center for New Energy & Intelligent Connected Vehicle, Wuhan University of Technology, Wuhan 430070, China; 4Institute of Advanced Materials and Manufacturing Technology, Wuhan University of Technology, Wuhan 430070, China; 5SAIC-GM-Wuling Automobile Co., Ltd., Liuzhou 545007, China

**Keywords:** plant–fiber composites, microcellular foam molding, life cycle assessment, digital twin, carbon neutrality

## Abstract

The development and utilization of new plant–fiber composite materials and microcellular foam molding processes for the manufacturing of automotive components are effective approaches when achieving the lightweight, low-carbon, and sustainable development of automobiles. However, current research in this field has mainly focused on component performance development and functional exploration, with a limited assessment of environmental performance, which fails to meet the requirements of the current green and sustainable development agenda. In this study, based on a life cycle assessment, the resource, and environmental impacts of plant–fiber composite material automotive components and microcellular foam molding processes were investigated. Furthermore, a combined approach to digital twinning and life cycle evaluation was proposed to conduct resource and environmental assessments and analysis. The research results indicate that under current technological conditions, resource and environmental issues associated with plant–fiber composite material automotive components are significantly higher than those of traditional material components, mainly due to differences in their early-stage processes and the consumption of electrical energy and chemical raw materials. It is noteworthy that electricity consumption is the largest influencing factor that causes environmental issues throughout the life cycle, especially accounting for more than 42% of indicators such as ozone depletion, fossil resource consumption, and carbon dioxide emissions. Additionally, the microcellular foam molding process can effectively reduce the environmental impact of products by approximately 15% and exhibits better overall environmental performance compared to chemical foaming. In future development, optimizing the forming process of plant–fiber composite materials, increasing the proportion of clean energy use, and promoting the adoption of microcellular foam injection molding processes could be crucial for the green and sustainable development of automotive components.

## 1. Introduction

There are many advantages of using fiber-reinforced composite materials for manufacturing automotive components, such as their lightweight, high strength, and high plasticity, which have become common materials in the manufacturing process of automotive parts [1,2,3]. Currently, fiber-reinforced composite materials prepared from carbon and glass fiber have the best performance and superb mechanical properties as critical materials in automotive components production. However, due to their complex production process, expensive, and high environmental impact, they are contrary to the current theme of carbon neutrality and low-carbon green development, which seriously limits their further large-scale application and promotion. Hence, there is an urgent need to develop novel, low-cost, eco-friendly fiber substitutes, and plant–fiber-reinforced composites are gradually coming to the forefront.

Plant fibers offer a wide range of options, including bamboo, cotton, hemp, and fruit shell fibers [4,5,6]. These natural fibers are abundant, widely sourced, renewable, and cost-effective, making them excellent alternatives to traditional fibers [7]. Polypropylene (PP) plays an extremely important role in the field of automotive materials as a wide range and large variety of general-purpose plastics with excellent performance and low cost are used in automotive manufacturing and the production of automotive components [8]. The preparation of plant–fiber-reinforced pp composite automotive components when using plant fibers as reinforcements to enhance product performance while controlling environmental impact is an effective way to achieve sustainable automotive development [9,10,11,12]. However, compared to traditional fiber materials, plant–fiber composites have a higher density, which significantly hinders the lightweight design and development of automobiles [13]. As a result, numerous experts and scholars have been continuously exploring new materials and processes to enhance the overall performance of plant–fiber composites [14], accelerating the green and sustainable development of the automotive industry [15,16,17,18].

Microcellular foam molding is an important polymer processing technology that allows for the formation of components with a fine porous structure after injection molding. This technology enhances the mechanical performance of the components and meets the requirements of lightweight design [19]. Microcellular foaming molding offers advantages, such as a unique structure, lightweight, and good sound, and thermal insulation properties [20,21]. It has broad application prospects in various fields, including automotive, construction, and aerospace [22,23]. Utilizing the microcellular foaming molding process to fabricate plant–fiber composite components enable a lightweight design while meeting the mechanical performance criteria [11]. Research has been conducted on various plant fibers such as jute fiber [24], crop straw fiber [25], sawdust, and wood pulp fiber [26,27]. Additionally, Qiao et al. [28] investigated the lightweight, soundproofing, and thermal insulation functions of plant–fiber foamed composite components through experimental studies. The development of plant–fiber foamed composite components aligns with the global theme of green development and holds significant importance for the sustainable development of the automotive industry.

However, the current research on plant–fiber composite foam molding has primarily focused on analyzing foam properties and their associated influencing factors, neglecting the analysis of the naturally renewable and environmentally friendly attributes of plant fibers. Moreover, there has been a significant research gap when conducting resource and environmental impact assessments for plant–fiber composite products; this has considerably hampered the further optimization and development of plant–fiber composite components in the realms of low-carbon manufacturing and sustainable development. To address this gap and foster the advancement of green and environmentally friendly materials, this study employed a life cycle assessment (LCA) [29,30,31] to investigate the low-carbon emission characteristics and resource and environmental impacts of plant–fiber composite automotive components and microcellular foaming molding processes. This study follows a comprehensive four-step approach, including goal and scope definition, inventory analysis, life cycle assessment, and the interpretation and explanation of results. In addition, this study incorporates subjective quantification methods and the techniques utilized by users to facilitate the efficient quantitative assessment of component life cycles. By utilizing the LCA methodology, this study aimed to provide essential theoretical support for the further optimization and development of plant–fiber composite components in the domain of low-carbon manufacturing and sustainable development. It not only explores the environmental performance of existing plant–fiber composite components in conjunction with microcellular foaming molding processes but also contributes to filling the research gap concerning resource and environmental impact assessments for plant–fiber composite products.

In addition, this study addresses challenges in relation to data availability, reliability, and system boundary definitions in the life cycle assessment (LCA) process. To overcome these challenges, this study incorporated digital twin technology to create a virtual representation of the entire life cycle of plant–fiber composite automotive components, specifically during the manufacturing stage. By utilizing the digital twin model, missing data could be estimated or predicted, facilitating a comprehensive life cycle assessment of the target subject. Digital twin technology is an advanced simulation technique that enables the virtual interconnection of intelligent manufacturing processes [32]. It leverages preliminary data analysis to predict behaviors and conduct virtual simulations of the manufacturing process, effectively addressing the data scarcity issues encountered in conventional LCA methodologies [33,34]. This integration of digital twin technology and LCA offers a novel and robust approach when evaluating the resource and environmental impacts associated with plant–fiber composite automotive components, enhancing the accuracy and reliability of sustainability assessments in the field of materials and manufacturing.

In conclusion, this study aimed to investigate the resource and environmental impacts of plant–fiber composite automotive components and microcellular foam molding processes by proposing an innovative approach that combines digital twin technology and life cycle assessments. This research sought to address current challenges in relation to the comprehensive assessment of resources and environmental implications throughout the entire life cycle of plant–fiber composite materials and microcellular foam molding processes. By integrating digital twin technology and life cycle assessments, this study provided an advanced methodology for analyzing the resource and environmental effects associated with plant–fiber composite automotive components and microcellular foam molding processes, contributing to the knowledge in the field of sustainable materials and manufacturing.

## 2. Materials and Methods

### 2.1. Goal and Scope Definition

The objective of this study was to establish a comprehensive lifecycle assessment model for automotive components made from plant–fiber composites, with a focus on investigating their environmental performance. Additionally, this study aimed to explore the emissions reduction potential in the microcellular foam injection molding process. Various control schemes were designed for both materials and processes, and the specific settings of this research plan were as follows.

Scenario 1: Chemical Foaming Injection Molding of Plant–Fiber-Reinforced Composite Materials. Automotive components are manufactured through the injection molding process using plant–fiber-reinforced modified PP materials, which undergo chemical foaming to create microcellular structures.

Scenario 2: Physical Foaming Injection Molding of Plant–Fiber-Reinforced Composite Materials. Automotive components are manufactured through the injection molding process using plant–fiber-reinforced modified PP materials, which undergo physical foaming to generate microcellular structures.

Scenario 3: Injection Molding of Plant–Fiber-Reinforced Composite Materials. Automotive components are directly manufactured through the injection molding process using plant fiber-reinforced modified PP materials.

Scenario 4: Conventional Injection Molding of Pure PP Materials. Automotive components are directly manufactured through the injection molding process using pure PP materials without the need for further processing.

Based on the proposed research plan, this study aimed to investigate the carbon emissions performance and environmental impact of plant–fiber composite materials in the foamed molding process of automotive components. This research encompassed the assessment of environmental performance during the production stage of plant–fiber composite automotive parts and a comparative analysis of the environmental effects of different types of foamed injection molding processes. The objective was to enhance an understanding of the resource utilization and environmental implications associated with the current production process of injection-molded automotive components, thereby providing technical support for future low-carbon, environmentally sustainable development. In this study, small and medium-sized automotive components were selected as reference samples, with the automotive seat angle adjuster wrench, previously manufactured by our research group, serving as the target component.

The system boundary of the LCA model for plant–fiber composite automotive components was established based on the LCA standard, as shown in Figure 1. With the system boundary of the LCA model established in this research, the resource and environmental impacts of the production and processing activities of plant–fiber composite automotive components from the “cradle” to the “gate” of the life cycle were discussed, including the extraction and processing of plant fiber, the preparation of plant–fiber-reinforced pp composites, and the microcellular foam injection molding of automotive components. In particular, the plant fiber extraction and processing phase, including energy and material consumption in the harvesting, transporting, crushing, and chemical treatment of crop straw to extract fibers, alongside the preparation phase of plant–fiber-reinforced PP composites, mainly considers the raw material and energy consumption in various composition ratios and composite masterbatch preparation processes under optimal mechanical performance conditions. It consists of a twin-screw extruder-based masterbatch preparation process for plant–fiber, PP-blended composite materials, and a twin-screw extrusion wire cutting and granulating process. It should be noted that in the above stage, the optimal conditions of mechanical properties were based on the previous studies of this group and are not repeated in this study. The microcellular foam injection molding phase of automotive components comprises mainly the material consumption and corresponding energy use involved in the microcellular foam molding process, which encompasses the manufacture of plant–fiber composite automotive components based on the microcellular foam molding process and the subsequent transportation and application of products.

In order to assess and clearly demonstrate the resource and environmental impacts of plant–fiber composite automotive components throughout their lifecycle, this study defines the functional unit as 1 kg of an automotive seat recliner wrench. The inventory data collected during the assessment process needed to be appropriately converted to the impacts associated with the production of 1 kg of the automotive seat recliner wrench in order to explore the resource and environmental impacts of different research scenarios during the product manufacturing process. Additionally, it should be noted that considering the high energy consumption during the start-up process of production equipment, this study utilized the average energy consumption of equipment for analysis. The three-dimensional model and relevant parameters of the automotive seat recliner wrench can be found in the supporting documentation. The parameters relating to the 3D model of the automotive seat adjuster wrench can be found in Figure A1 and Figure A2, while the specific model can be found in the additional model file.

### 2.2. Life Cycle Inventory Analysis

Lifecycle inventory data analysis is the most critical aspect of the Life Cycle Assessment (LCA) [35]. The quality of inventory data determines the value of the assessment results. In this study, the lifecycle inventory data were divided into background data and foreground data [36]. The background data were sourced from the China Lifecycle Database (CLCD) and the Ecoinvent database built into Simapro software, which is primarily used for basic energy and material data. The foreground data for the processing of plant–fiber composites and microcellular foaming injection molding stages were mainly derived from the previous experimental records of our research group, publicly available data from research papers published by experts in relevant disciplines, the manufacturers of experimental equipment, statistical yearbooks published by local governments, and data obtained through collaboration with partner enterprises. The relevant inventory data for each assessment scheme under 1 functional unit are shown in Table 1.

In addition, this study addressed the issue of multi-physics field coupling during the microcellular foaming injection molding stage of plant–fiber composites by employing a digital twin-based modeling and analysis approach. The process is depicted in Figure 2. Diverging from the limitations of acquiring personalized performance data through the conventional means of obtaining lifecycle assessment data, digital twin technology was utilized to construct a digital twin model based on generic data. Through continuous personalized iteration and optimization, the digital twin model was endowed with the capability to handle specific operating conditions, thereby facilitating the acquisition of required data for a lifecycle assessment under complex operating conditions.

This study’s life cycle inventory data were carefully organized and partially adjusted for the above data sources to make the assessment results closer to the actual resource and environmental impacts. The above data were adjusted for their actual development in the Chinese region, including energy, electricity, and essential materials. Moreover, a data acquisition method combining theoretical equation derivation and DT simulation analysis was proposed in this study for complex working conditions in the injection molding phase. The inventory data analysis for each step in the life cycle of plant–fiber composite automotive components is shown below.

#### 2.2.1. Extraction and Processing Stages of Plant Fiber

An LCA inventory data analysis method was used to divide the plant fiber extraction phase into four steps: crop straw harvesting and baling, straw crushing and processing, alkaline solution soaking and extraction, and high-speed rotary screening. While most of these steps involved energy and power consumption, only the extraction process of fiber soaking under alkaline solution conditions involved physicochemical reactions and the input and generation data of the relevant reactions needed to be clarified. For the extraction of fibers, soaking in an alkaline solution of sodium hydroxide was used in this study to remove other impurities in the separated plant straw [37,38], followed by fiber separation using the rotary sieving method to obtain more intact plant fibers.

#### 2.2.2. Preparation Stage of Masterbatches for Plant–Fiber-Reinforced PP Composites

This phase’s life cycle inventory data were mainly used for preparing the plant–fiber-reinforced PP composites. Considering that the subject of this study was a conventional small part with simple material ratios and no mass production was required, the masterbatch preparation step was ignored, and the plant–fiber composites were directly modulated with suitable ratios. Among them, different control scenarios were set up for the material ratio and processing of plant–fiber composites, which included the use of pure PP material based on the traditional injection molding process to prepare the products, the use of plant–fiber composites based on the conventional injection molding process to prepare the products, and the use of plant–fiber composites based on the microcellular foam molding process to prepare the products. The PP, coupling agent, and plant fiber mass ratio in the plant–fiber composites were 67:3:30 [39] based on the foundation of this group and other previous studies in this field [40,41], which were obtained by twin-screw extrusion mechanical drawing and granulation to acquire the plant–fiber composite pellets for backup.

#### 2.2.3. Microcellular Foam Injection Molding Stage for Automotive Parts

The injection molding phase of plant–fiber composites is a crucial part of the LCA process. This study used the HDX 128 (Ningbo Haida, Ningbo, China) single-screw injection molding machine to complete the manufacturing of automotive components. In addition, this study had several experimental scenarios, including the injection molding of plant–fiber composite automotive parts using chemical foaming agents, the physical foaming of plant–fiber composite automotive parts using supercritical gases, and conventional non-foaming plant–fiber composite automotive components. AC and ZnO were used as chemical foaming agents, and physical foaming was performed with supercritical N_2_ in this study. Furthermore, aiming at the problem of the difficult data acquisition of multi-physical field coupling conditions in the injection molding process of plant–fiber composites, this study built a DT system architecture for the microcellular foam injection molding phase of plant–fiber composites based on the above foundation, as shown in Figure 3. The multi-level combination of the service layer to clarify the service targets and the goals of the DT model, the data layer to determine the data type within the model, and the connection layer to build an intelligent information connection framework, realized the virtual-to-real conversion of the physical and virtual layers. Hence, a DT model of the microcellular foam injection molding process of plant–fiber composites was constructed with the function of simulation, parameter optimization, and data acquisition to meet the data usage requirements for the LCA of the injection molding phase of plant–fiber composites.

While analyzing the DT model establishment process through a combination of theoretical derivation and practice, the injection molding process of plant–fiber composites was profiled as follows by combining preliminary data with the previous research base of our research group [42]. The energy and power consumption were normalized in the injection molding phase of the plant–fiber composites for the convenience of this study, while the injection molding process was decomposed into four steps: the melt filling phase, the holding pressure phase, the plasticizing phase, and the filling phase of the composites. Therefore, mathematical models were established for the above steps separately for analysis, combined with theoretical mechanics, thermodynamics, and fluid dynamics analysis. The theoretical model of energy consumption *E_injection_* in the melt filling stage of the microcellular foam injection molding process was determined under ideal conditions as follows:(1)$$E_{injection}=\int_{0}^{t_{1}}(F_{\eta }+F_{f}+P_{injection}A_{1})V_{z}dt+m(V_{2}^{2}-V_{1}^{2})/2$$
where *t*_1_ is the time required for the filling phase, s; $F_{\eta }$ refers to the viscous resistance of the melt during the screw movement, N; $F_{f}$ represents the material frictional resistance during the screw movement, N; *P_injection_* refers to the injection pressure, Pa; $A_{1}$ is the cross-sectional area of the screw, m^2^; *V_z_* is the speed of the melt moving relative to the screw during the filling process, m/s; *m* represents the mass of the screw, kg; *V*_1_ refers to the speed of the screw at the beginning of the filling, m/s; *V*_2_ is the speed of the screw at the end of the filling, m/s.

The pressure holding phase had significance for product forming and quality, and this phase was analyzed in the ideal state for energy consumption *E_pack_* theory model as follows.
(2)$$E_{pack}=\int_{0}^{t_{2}}\left[(F_{\eta }^{\prime }+F_{f}+P_{pack}A_{1})(1-\delta )H_{total}/t_{2}A_{1}\right]dt$$
where *t*_2_ is the duration of the holding pressure stage, s; the viscous resistance of the screw during the holding pressure phase after the conversion of the screw by the V/P point is denoted as $F_{\eta }^{\prime }$, N; $P_{pack}$ refers to the pressure in the mold cavity during the holding pressure phase, Pa; δ indicates the V/P conversion point divided by the filling volume, and *H_total_* refers to the total volume of the cavity and runner, m^3^.

The plasticization phase included thermal and mechanical energy consumption. This process was based on the law of energy conservation and the heat balance equation for material melting and compaction. The phase was analyzed in the ideal energy consumption *E_melt_* theoretical model.
(3)$$\begin{array}{c} E_{melt}=G_{f}[(T_{o}-T_{s})C_{p}+H_{f}]t_{3}+P_{h}G_{f}t_{3}/\rho_{m}\\ +C_{o}\varepsilon [{\left(T_{o}/100\right)}^{4}-{\left(T_{s}/100\right)}^{4}]A_{k}t_{3}+(T_{o}-T_{s})\lambda \pi (R_{2}^{2}-R_{1}^{2})t_{3}/l_{b} \end{array}$$

It contained the heat energy, potential energy, and heat radiation loss of the material in the plasticizing phase; *G_f_* indicates the mass flow rate, kg/s; *T_s_* refers to the initial temperature of the plasticizing degree, °C; *T_o_* is the final temperature of the plasticizing phase, °C; *C_p_* refers to the specific heat capacity of the material, J/(kg-°C); *H_f_* indicates the latent heat of melting of the material; *t*_3_ refers to the plasticizing time, s; *P_h_* refers to the back pressure set in the plasticizing phase, MPa; *ρ_m_* refers to the density of the melt; *C_o_* indicates the blackbody radiation coefficient; ε is the blackness of the barrel material; *A_k_* is the surface area of the barrel for the injection molding machine, m^2^; λ indicates the heat transfer coefficient of the barrel material, W/(m-K); *R*_1_ refers to the inner radius of the barrel, m; *R*_2_ indicates the outer radius of the barrel, m; *l_b_* is the overall length of the barrel, m.

Microcellular foam molding is vital for components to reduce their quality and improve performance. There are two types of microcellular foam molding processes which can be classified as physical foam molding and chemical foam molding according to the foaming method. Among them, chemical foaming is based on gas generation, which is induced by temperature changes in the foaming agent during the injection molding process, which results in a porous structure inside the components. It is a process that is more dependent on material properties and has less impact on energy consumption and emissions; therefore, it is not discussed in depth here. However, the physical foam molding process requires the supercritical filling of inert gas during the plasticization phase to create micro-pores inside the injection molded components. While in the gas injection phase, the key is to pressurize inert gas in the cylinder to supercritical conditions for filling; this consumes more energy. To investigate the energy consumption of this process, this study determined the theoretical model of energy consumption in the air injection phase of the foam injection molding process as follows.
(4)$$\begin{array}{c} E_{gas}=[p_{s1}Q_{l1}\ln[p_{d1}(X_{s1}+X_{d1})/2p_{s1}X_{s1}]+\\ p_{s2}Q_{l2}\ln[p_{d2}(X_{s2}+X_{d2})/2p_{s2}X_{s2}]]m_{total}C_{gas}/\rho_{gas}Q_{l2} \end{array}$$

In the theoretical model of the energy consumption for the gas injection phase, *p_s_*_1_ and *p_s_*_2_, respectively, are the inlet pressure values of the primary/and secondary compressors, MPa; *Q_l1_* and *Q_l2_*, respectively, are the actual discharge volumes of primary and secondary compression, m^3^; *p_d_*_1_ and *p_d_*_2_, respectively, are the discharge pressure values of the primary and secondary compressors, MPa. *X_s_*_1_ and *X_d_*_1_ refer to the inlet and exhaust gas compression coefficients for primary compression; *X_s_*_2_ and *X_d_*_2_ refer to the inlet and exhaust gas compression coefficients for secondary compression; *m_total_* indicates the total mass of the injection molded components and runners, kg; *C_gas_* represents the initial gas concentration; *ρ_gas_* refers to the density of supercritical gas, kg/m^3^.

In addition to the above analysis of energy consumption in each phase of injection molding for plant–fiber composite automotive components, the injection mold was also something we needed to consider. It is essential that the energy consumption of the manufacturing process for the automotive seat adjuster wrench mold analyzed in this study should also be considered in the system boundary. Therefore, it was easy to find the energy consumption *E_moulding_* of the mold for producing functional units of components by taking the number of components injected as the life unit in this study [43]. Hence, combining the above practical and theoretical analysis, a DT model of the injection molding process of plant–fiber composites was established by combining the existing injection molding method with DT technology, and the interface effect is shown in Figure A3.

### 2.3. Life Cycle Impact Assessment

This study conducted a cradle-to-gate resource and environmental impact assessment on plant–fiber composite automotive components based on the LCA method under ISO 14040. The research process set up the control scenarios, including the pure PP material automotive components, plant–fiber composite automotive components, and microporous foam plant–fiber composite automotive components. The LCA model based on Simapro 9.0 was established and used the ReCiPe 2016 method to evaluate the resource and environmental impacts of the products under different scenarios, covering carbon emissions analysis including IPCC (GWP 100a) and environmental impacts including atmosphere, soil, and waters. The assessment results were normalized, and sensitivity analysis was conducted to verify the accuracy and reliability of the assessment results for significant influencing factors.

### 2.4. Sensitivity Analysis

The sensitivity analysis was introduced for critical data in the product LCA process to quantify the fluctuation of results due to data variations [30,44,45]. Through the ±10% adjustment of the above critical parameter values, followed by the analysis with the fluctuation of product LCA results, this study also introduced the Monte Carlo analysis for uncertainty analysis and model reliability validation.

## 3. Results and Discussion

### 3.1. The Results of Environmental Impact Indicator Characterization

Environmental issues due to energy and material consumption during the injection molding of automotive components are critical when assessing the eco-sustainability of automobiles today. This study investigated the environmental impacts with four scenarios based on the ReCiPe midpoint method, which was divided into toxicity analysis, ozone impact analysis, eutrophication analysis, and resource impact analysis.

#### 3.1.1. Ecotoxicity Depletion

The results of the toxicity study analysis of the injection molding manufacturing process of plant–fiber composite automotive components are shown in Figure 4. From a macroscopic point of view, the impact of Terrestrial ecotoxicity and Human non-carcinogenic toxicity caused during the life cycle of several scenarios that were set up showed the most prominent performance, much higher than the three indicators, including Freshwater ecotoxicity, Marine ecotoxicity, and Human carcinogenic toxicity. On the other hand, electricity, energy consumption, and chemical reagents were the main contributors to toxicity impacts, which made the emissions of scenario 1 and scenario 2 almost the same under the toxicity analysis index, even if the use of chemical blowing agents led to higher emissions of scenario 1; however, both were lower than those of scenario 3. Therefore, the analysis showed that the microcellular foam injection molding process has the effect of reducing environmental toxicity impacts. Moreover, we found that the toxic emissions of these scenarios containing plant–fiber composites were much higher than those of pure PP materials regardless of the assessment measure, which meant that the application of plant–fiber composites increased toxic environmental emissions. The comparative analysis found that the reason for this was that the process of plant fiber extraction and the preparation of plant–fiber composites increased the loss of materials and energy, for which the environmental impact was higher than that of the conventional injection molding of pure PP materials. It is clear that in the future development of plant–fiber composites, developing new environmentally friendly processes is the premise that is critical to achieving the sustainable development of plant–fiber composites.

#### 3.1.2. Ozone Depletion

As shown in Figure 5a–c, in a comprehensive analysis of the ozone impact indicators of the Human health ozone formation potential (HOFP), Terrestrial ecosystems ozone formation potential (EOFP), and ozone depleting potential (ODP), it was found that the life cycle ozone impacted emissions from the automotive components made with plant–fiber composites, which were still much higher than those from conventional pure PP materials. Meanwhile, a comparative analysis of the three environmental indicators in the ozone impact analysis found that pollutant emissions under the HOFP and EOFP indicators were three orders of magnitude higher than the ODP. On the other hand, among all energy and materials, diesel, sodium hydroxide, and electricity were the most critical contributors to pollutant emissions, while electricity, in particular, accounted for 30–50% of overall emissions. This shows the strong correlation between electrical energy and ozone protection, which is why China should accelerate its transition to a cleaner power system in the future, increasing the proportion of clean electricity and reducing coal-fired power generation to reduce the pollution caused by the power generation process. Moreover, it was found that pollutant emissions were similar when comparing HOFP and EOFP, which showed that terrestrial ecosystems could be closely related to human health. Finally, a comparative analysis of the different study protocols showed the importance of the microcellular foam injection molding process in reducing pollutant emissions. As well as in response to the impact of the ozone, chemical foam injection molding had a better environmental performance than physical foam injection molding. This was because the supercritical gas required for physical foaming required multi-stage pressurization in the preparation process, which required a multi-stage compression motor, resulting in higher electrical energy consumption and impact on the overall process. Naturally, this is using the power of central China as a reference, which means that different countries and different regions could have different results concerning power conditions.

#### 3.1.3. Eutrophication Depletion

The prevention of eutrophication in waters is an important initiative to protect waters. In this study, the impact of water caused by the manufacturing process of plant–fiber composite automotive components was evaluated, and the results are shown in Figure 6a,b. The pollution of water caused by the manufacturing process of plant–fiber composite automotive components was assessed by analyzing the nitrogen and phosphorus content in the waters under the water body eutrophication assessment index. Above all, in the analysis of eutrophication against phosphorus, it was found that the overall impact of scenario 1 and scenario 2 on plant–fiber composite automotive components with the microcellular foam injection molding process was lower than that of direct injection molding in scenario 3 but higher than that of scenario 4 with the injection molding of pure PP material. On the other hand, the higher power consumption of the physical foam injection molding process led to a slightly lower impact caused by chemical foam injection molding in scenario 1 than in scenario 2. Conversely, the analysis of the eutrophication of nitrogen at this point found that the eutrophication impact produced by scenario1 was significantly higher than that of scenario 2 because of the large number of nitrogenous emissions produced by the AC foaming additives used in the chemical foaming injection molding process, thus leading to a larger eutrophication impact and even exceeding that of scenario3 which did not use the foaming process. Meanwhile, straw crops also became the main influencing factor in the analysis of nitrogen eutrophication.

#### 3.1.4. Resource Depletion

The resource impacts of the manufacturing and molding process of plant–fiber composite automotive components were mainly on land, petroleum, mineral, and water resources. This study found that the resource impact was closely related to material consumption in the product manufacturing process, where the impact of land and petroleum resources is shown in Figure 7a,b. Diesel and rice straw were the major contributors to land resource impacts. By contrast, electricity, chemical raw materials, and PP materials were the main contributors to the impact of petroleum. The principle of the supply cycle was followed for all the above impacts, and the results were clear.

In addition, as shown in Figure 7c,d, for the mineral resources impact analysis, it was found that the impact factors were mainly concentrated in petroleum resources, while the water resources impact was focused on the production process of several materials with high water consumption. It showed the resource impact of each link in the manufacturing process of plant–fiber composite automotive components and provided some references for adjusting the process and improving the technical solutions.

### 3.2. Climate Change

Lightweight and eco-carbon concepts have consistently been implemented in the manufacturing process of automotive components. The preparation of automotive parts based on the microcellular foam technology injection molding of plant–fiber composites is a bold attempt when integrating lightweight design and eco-carbon concepts. Therefore, to investigate further the environmental protection and low carbon performance of plant–fiber composites and microcellular foam techniques, this study evaluated the carbon emissions of the four scenarios set out above based on the IPCC (GWP 100a) assessment method. This is significant for the low-carbon development of automotive components and the eco-friendly and sustainable development of automobiles in the context of global “carbon neutrality”.

The results of the carbon emission assessment when comparing the manufacturing process of plant–fiber composite automotive components under different scenarios are shown in Figure 8. The area of the circle represents the carbon emission intensity of the segment, and different colors are used to distinguish the types of materials in different segments. In addition, a cut-off principle was used in this study to show the carbon emissions for less than 1% of the segments to express carbon emissions under various scenarios clearly and concisely.

Overall, the carbon footprint of the manufacturing process of plant–fiber composite automotive components was significantly higher than that of traditional pure PP material automotive components. The carbon footprint was about 6–7 times higher than the conventional pure PP material injection molding manufacturing process. This could be attributed to the addition of plant fiber extraction, processing, and preparing plant–fiber-reinforced PP composites compared with traditional pure PP materials. Meanwhile, further observation of the plant fiber extraction and composite material preparation processes from Figure 8 revealed that scenarios 1–3 had very similar carbon emission trends, which are formed a carbon emissions footprint with a path from plant fiber extraction to composite material preparation and then to product injection molding. The plant fiber extraction process was the most prominent carbon emitter. The chemical reagents used in this process, energy consumption, and electrical energy consumption were more than 70% of the total carbon emissions of the scenario in that location. Conversely, scenario 4 had a wide range of PP material sources alongside the current PP extraction and upstream preparation processes which are well-developed and mature; therefore, this scenario had a significantly lower carbon footprint.

The continuous in-depth comparative investigation found that the microcellular foaming process could effectively reduce carbon emissions by about 15% while reducing product quality and improving product performance. As for the microcellular foam molding process, the implementation of two methods within the investigation found that chemical foam molding had a better low-carbon performance than physical foam molding. As a result, the physical foaming process consumed supercritical gases that consumed more energy in the pressurized preparation process, leading to higher overall carbon emissions in the physical foaming and molding process. Moreover, the analysis from the perspective of the influence of a single factor found that electricity resources were the most significant contributor to carbon emissions in the manufacturing process of automotive components. Of the four scenarios set up in this study, the carbon emissions caused by the consumption of electric power resources accounted for 49%, 51%, 48%, and 59% of the respective total carbon emissions, indicating the importance of vigorously developing and using clean energy in the process of energy conservation and emissions reduction.

### 3.3. The Results of Sensitivity Analysis

This study analyzed carbon emissions in plant–fiber composite automotive component manufacturing processes. The sensitivity analysis results through critical parameters of electricity and plant fiber parameters are shown in Figure 9 and verify the uncertainty and accuracy of the assessment model. Plant fibers were significantly more sensitive than electricity and were the main contributor to carbon emissions in the production of the product, which is consistent with the results of the above investigation. On the other hand, using the microcellular foam injection molding process could effectively reduce the sensitivity of parameters, indicating that this process can effectively reduce product carbon emissions. Moreover, it could be concluded that the carbon emission control of the product could be started with plant fiber extraction and power energy consumption control. Improvements to the fiber extraction process, reductions in carbon emissions for the extraction process, optimizing the energy structure, and reducing the use of electricity and energy could all effectively reduce carbon emissions.

As a response to the multi-parameter uncertainty problem in the evaluation process, this study simulated the four scenarios of the study setup based on Monte Carlo analysis, and the results obeyed the Gaussian distribution completely. The relevant parameters are shown in Table 2. The data results show that the error between the actual evaluation values and the simulated values for the four research scenarios set up in this paper was at most 1% in 1000 simulations, which verified the accuracy of the evaluation analysis model established in this study.

## 4. Conclusions

This study developed an environmental and resource assessment model for the microcellular foam injection molding of plant–fiber composite materials using LCA. This study employed digital twinning technology to enhance the quality of inventory data and provide a comprehensive evaluation of the production process. The results reveal that compared to pure PP materials, plant–fiber composite automotive components can exhibit significantly higher resource consumption and environmental impacts, which are amplified nearly fivefold. However, the adoption of microcellular foam molding techniques, particularly chemical foaming, reduced the resource consumption and environmental impacts associated with automotive components by approximately 15%. The sensitivity and uncertainty analyses identified pre-treatment processes, electricity consumption, and chemical raw material usage as the primary factors influencing environmental impacts. Notably, electricity consumption was found to contribute over 42% to indicators such as ozone depletion, fossil resource consumption, and carbon dioxide emissions. These findings emphasize the importance of promoting clean energy development and optimizing energy structures in order to mitigate environmental impacts.

Based on the aforementioned analysis, to further drive the development of eco-friendly automotive components and promote sustainable growth in the industry, the future focus on new plant–fiber composite materials for car parts should consider the following aspects. Firstly, the research and development of pre-processing and manufacturing techniques should be prioritized for plant–fiber composites, aiming for green and efficient fiber extraction and high-performance material shaping processes to achieve environmentally friendly manufacturing. Secondly, the widespread use of microcellular foam injection molding should be promoted in the automotive sector as an effective approach for lightweight vehicles and green development. Lastly, efforts to improve the electricity infrastructure in manufacturing should be accelerated, encourage the use of clean energy, and optimize energy sources to reduce environmental impacts during production.

## Figures and Tables

**Figure 1 materials-16-04952-f001:**
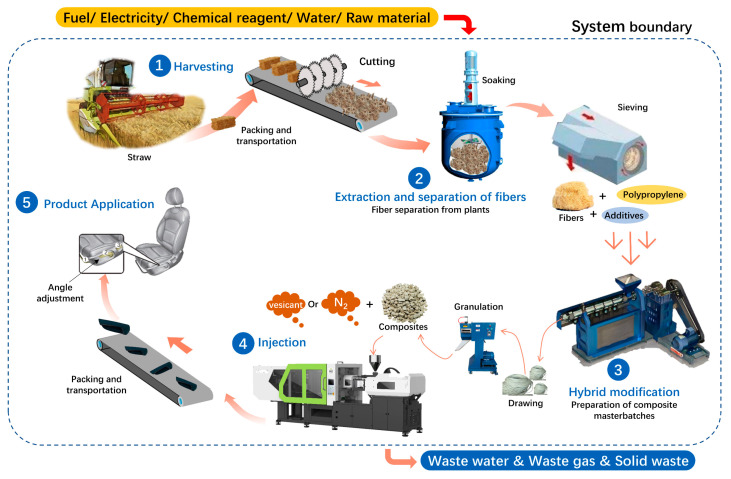
The boundary of the LCA system for plant–fiber composite automotive components.

**Figure 2 materials-16-04952-f002:**
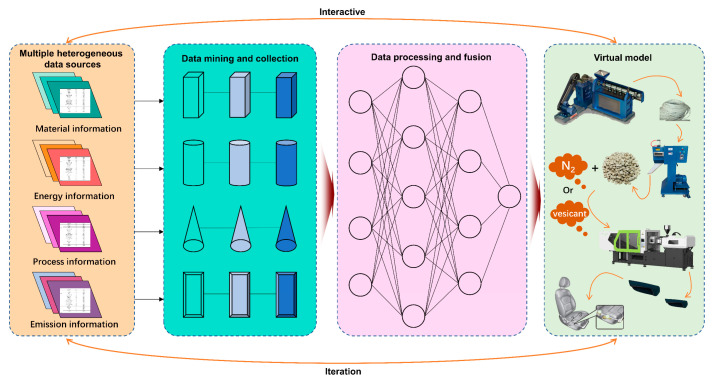
The process of DT data modeling management.

**Figure 3 materials-16-04952-f003:**
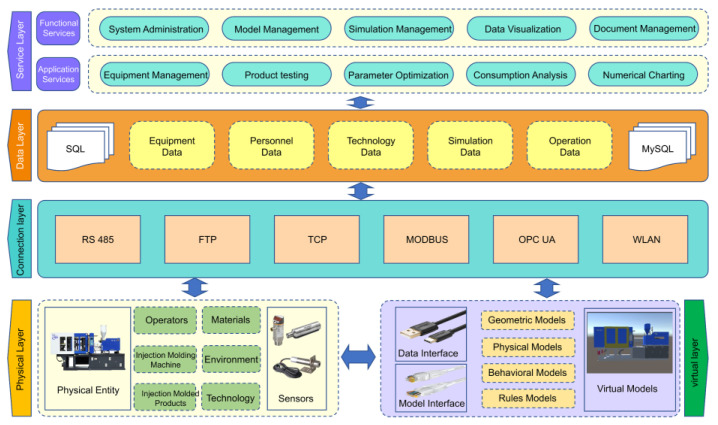
The architecture of the DT data modeling system.

**Figure 4 materials-16-04952-f004:**
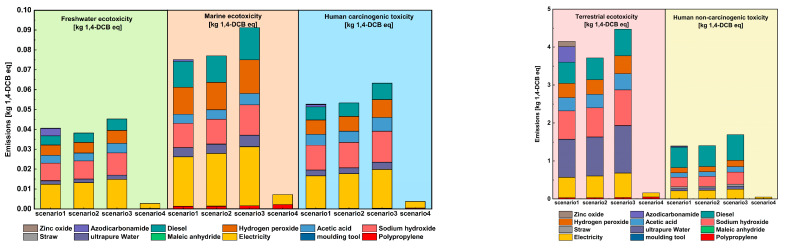
Comparison of toxic impacts analysis.

**Figure 5 materials-16-04952-f005:**
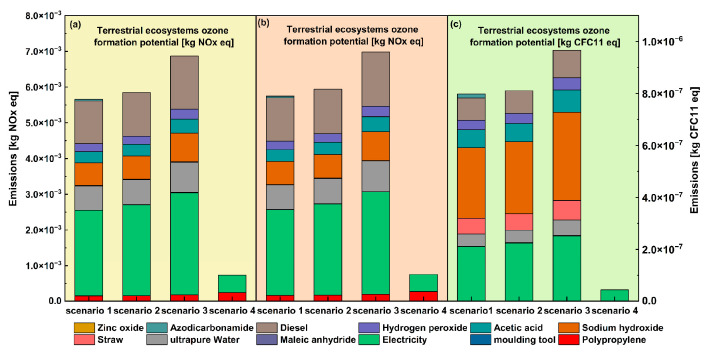
Comparison of ozone impact analysis.

**Figure 6 materials-16-04952-f006:**
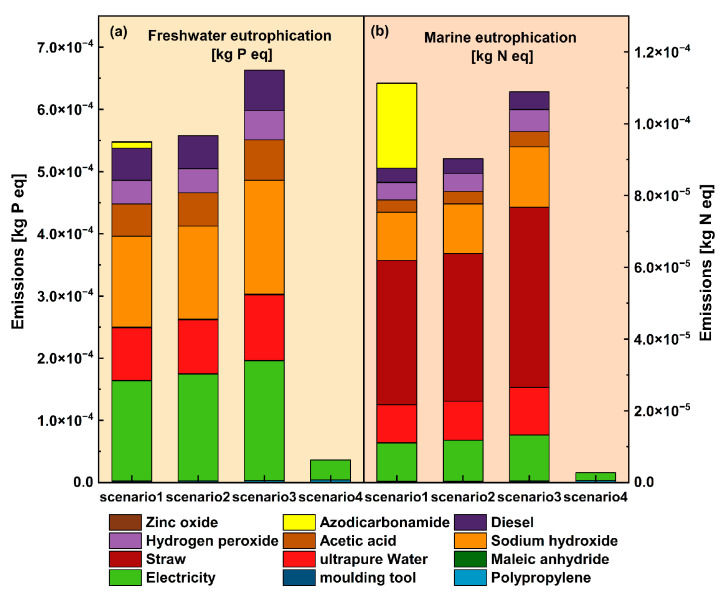
Comparison of eutrophication impact analysis.

**Figure 7 materials-16-04952-f007:**
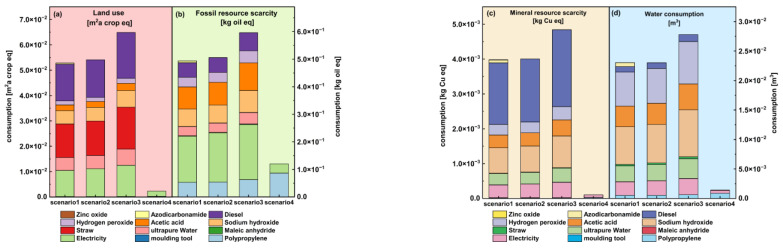
Comparison of resource impact analysis.

**Figure 8 materials-16-04952-f008:**
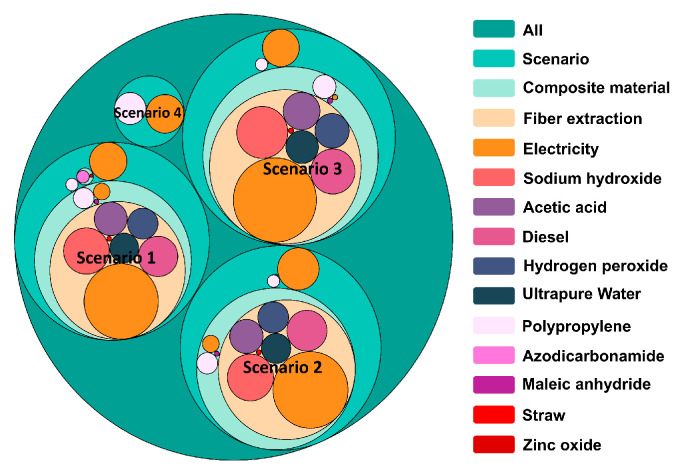
Comparison of carbon emissions for plant–fiber composite automotive components with multiple scenarios of the molding process.

**Figure 9 materials-16-04952-f009:**
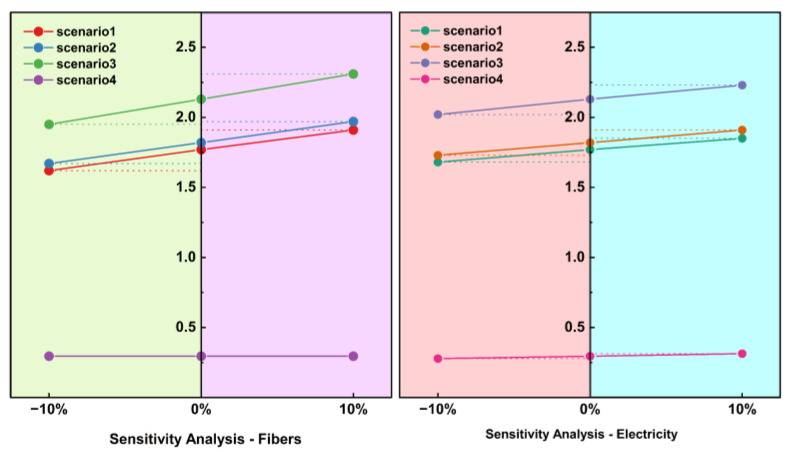
Sensitivity analysis of fiber content and electricity consumption.

**Table 1 materials-16-04952-t001:** Primary inventory data under functional unit conditions.

	Parameter Name	Amount	Unit
scenario 1	Electricity	5.13	kWh
	plant–fiber composite	0.80	kg
	Water	2.00	kg
	Polypropylene	0.20	kg
	Azodicarbonamide	0.03	kg
	Zinc oxide	0.05	kg
	Molding tool	5.26 × 10^−6^	p
scenario 2	Electricity	6.32	kWh
	plant–fiber composite	0.82	kg
	Water	2.00	kg
	Polypropylene	0.20	kg
	Nitrogen	0.02	kg
	Molding tool	5.26 × 10^−6^	p
scenario 3	Electricity	4.98	kWh
	plant–fiber composite	1.00	kg
	Water	2.00	kg
	Polypropylene	0.20	kg
	Molding tool	5.26 × 10^−6^	p
scenario 4	Electricity	5.24	kWh
	Water	4.00	kg
	Polypropylene	1.20	kg
	Molding tool	5.26 × 10^−6^	p

**Table 2 materials-16-04952-t002:** Statistical evaluation of output distribution.

Parameter Name	Scenario 1	Scenario 2	Scenario 3	Scenario 4
Trials	1000	1000	1000	1000
Base case	1.77	1.82	2.13	0.295
Mean	1.78	1.84	2.11	0.295
Median	1.75	1.80	2.11	0.292
Standard deviation	0.166	0.168	0.195	0.0301
Standard error of mean	5.26 × 10^−3^	5.32 × 10^−3^	6.18 × 10^−3^	9.52 × 10^−4^
Skewness	0.934	0.973	0.702	0.858
Coefficient of variability	9.39%	9.26%	9.18%	10.2%
Minimum	1.41	1.43	1.6	0.22
Maximum	2.42	2.48	3.1	0.431

## Data Availability

Not applicable.

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
