# Peer review of "Unveiling Sustainable Potential: A Life Cycle Assessment of Plant–Fiber Composite Microcellular Foam Molded Automotive Components"

_materials, 2023, doi:10.3390/ma16144952_

Round 1

Reviewer 1 Report

This paper focuses on the development and utilization of new plant fiber composite materials and microcellular foam molding processes for manufacturing automotive components, with an aim to achieve lightweight, low-carbon, and sustainable development of automobiles. It acknowledges a gap in current research that emphasizes component performance and functional exploration while overlooking environmental performance assessment. The findings reveal that plant fiber composite materials have higher resource and environmental impacts compared to traditional materials, primarily due to disparities in early-stage processes and the consumption of electrical energy and chemical raw materials. Additionally, the study demonstrates that the microcellular foam molding process exhibits superior overall environmental performance and can reduce product environmental impact by approximately 15% compared to chemical foaming.

The paper is well-structured and presents interesting results. However, there are concerns that should be addressed before considering it for publication. Firstly, the quality of all figures needs improvement. The poor resolution and dimensions make the images, text, and labels unreadable. Furthermore, the authors should clarify their original contribution to the field, as several similar papers have already been published in the literature. It is recommended to clearly indicate the chosen experimental conditions (mold geometry, molding machine, material properties, etc.), the target product considered (surface quality, thickness, voids, fixtures, etc.), and how these choices may impact the results. Discussing the potential influence of alternative molding conditions or different parts would broaden the scope of the work and provide valuable insights for a more comprehensive discussion.

Author Response

Dear Reviewer,

Greetings! Thank you for reviewing our submitted paper and providing valuable feedback. According to your suggestions, we have made the necessary revisions to the manuscript. The detailed changes and responses are compiled in the attached document for your reference.

We sincerely appreciate your review, as your comments have significantly improved the quality of our research and paper. Should you have any further requests or additional feedback, please do not hesitate to let us know.

Best regards,

Wei Guo

Reviewer 2 Report

The manuscript is dedicated to an interesting topic – a comparison of the LCA of an automotive part made of neat PP, PP filled with plant fibers, and 3 types of foamed PP with plant fibers. The research is well structured and the results are clearly presented. However, some recommendations are given below:

1. The figures in the manuscript are of very poor quality and it is close to impossible to read them.

2. Table 1, Scenario 4, PP amount: are you sure it is 12 kg?

3. Line 329: “Microcellular foam molding is vital for components to reduce quality and improve 329 performance.”  Are you sure you wanted to write “reduce quality”?

4. It would be interesting to see the mechanical performance of the examined car part made of different materials (neat PP, filled PP, and foamed filled PP). It can turn out that the mechanical performance of foamed PP filled with plant fibers is much higher than those of neat PP, and the part is overengineered and does not really need filling with plant fibers. In this case, the comparison of the environmental impact of the examined car parts made of the researched materials little bit loses its relevancy, as there is no point to compare parts that are not equivalent by their mechanical performance. Could you please clarify this issue?

5. Lines 34-37: “In future development, optimizing the forming process of plant fiber 34 composite materials, increasing the proportion of clean energy use, and promoting the adoption of 35 microcellular foam injection molding processes are crucial for the green and sustainable develop- 36 ment of automotive components in the future.” Is 2 times “in future” really needed in this sentence?

6. As the authors examined PP composites in their study, it is recommended to improve the introduction section with a short overview of the properties of PP composites, which can be found: https://doi.org/10.1007/978-3-030-12903-3_9

Author Response

(The authors gave the same response as above.)

Round 2

Reviewer 1 Report

I recommend the publication of this document.

Reviewer 2 Report

The authors corrected the manuscript according to my recommendations